# Speedy bearings to slacked steering: Mapping the navigation patterns and motions of Viking voyages

**Peter Takacs**[1], **Denes Szaz**[1], **Adam Pereszlenyi**[1,2], **Gabor Horvath**[1] *

1 Department of Biological Physics, ELTE Eötvös Loránd University, Budapest, Hungary, 2 Deutsches Meeresmuseum, Stralsund, Germany

☉ These authors contributed equally to this work.
* gh@arago.elte.hu

**Data Availability Statement:** All relevant data are within the paper.

**Funding:** DS was supported by the Hungarian UNKP-21-4 New National Excellence Program of

## Abstract

Viking sailors ruled the North Atlantic Ocean for about three hundred years. Their main sailing route was the 60˚ 21' 55'' latitude between Norway and Greenland. Although they did not have a magnetic compass, in sunshine they used a sun-compass to determine the geographical north (solar Viking navigation: SVN). It has been hypothesized that when the Sun was invisible, Viking navigators determined the direction of polarization of skylight with sunstones (dichroic/birefringent crystals), and then estimated the geographical north using the sun-compass (sky-polarimetric Viking navigation: SPVN). Many details of the hypothetical SPVN have been thoroughly revealed in psychophysical laboratory and planetarium experiments. Combining these results with measured celestial polarization patterns, the success of SPVN was obtained as functions of sailing, meteorological and navigation parameters (sunstone type, sailing date, navigation periodicity, night sailing, cloudiness conditions). What was so far lacking in this experimental and computational archeological approach is the study of the success of SVN and a combined navigation using solar cues in sunshine (SVN) and sky polarization at invisible Sun (SPVN), the latter being the most realistic method. In this work we determine the success of the sole SVN and the combined SVN-SPVN relative to the mere SPVN for three navigator types (determining the intended sailing direction with large, medium or small frequencies) at spring equinox and summer solstice, with and without night sailing. We found that to maximize the sailing success, navigators had to choose different navigation methods depending on the navigation frequency. Using sky polarization with very frequent navigation, resulted in the highest chance to survive a three-week voyage from Norway to Greenland.

## 1. Introduction

Viking sailors ruled the North Atlantic Ocean for about three hundred years (between *Anno Domini* 900 and 1200) [1]. Among others, they founded settlements in south Greenland and Newfoundland [2]. Their main sailing route was along the northern latitude of 60˚ 21' 55'' between Bergen (middle Norway) and Hwarf (south Greenland) [3].

the Ministry for Innovation and Technology from the source of the National Research, Development and Innovation Office. This research was also supported by a KDP-2020-ELTE-1010099 fellowship/grant from the National Research, Development and Innovation Office to PT, who received a further financial support from the Doctoral School of the Physical Institute of the Eötvös Loránd University. The funders had no role in study design, data collection and analysis, decision to publish, or preparation of the manuscript.

Since Viking sailors did not have a magnetic compass [4], they should have to use other methods for navigation during their sea voyages [5–7]. It is an archeologically corroborated fact that Viking navigators used a sun-compass by which they could determine the direction of the geographical north in sunshine [8–10]. This orientation method is called 'solar Viking navigation' (SVN). It has been hypothesized that when the Sun was not visible (because it was below the horizon or hidden by clouds) Vikings determined the direction of polarization of skylight with sunstones, and then estimated the geographical north using the sun-compass [11–13]. This hypothetical orientation method is named 'sky-polarimetric Viking navigation' (SPVN). According to the hypothesis, sunstones might have been dichroic or birefringent crystals, such as tourmaline, cordierite or calcite, functioning similarly as linear polarization analyzers [14–17]. We emphasize, however, that in spite of the wide-spread hypothesis of SPVN, it is not based upon archeological artefacts or written documents.

The SPVN has four steps [18–20] (Fig 1): 1) Determination of orientations O1 and O2 perpendicular to the directions of polarization of skylight in two celestial points C1 and C2 with the use of sunstones (Fig 1A). 2) Determination of the position P of invisible (cloud-occluded) Sun coinciding with the intersection P of the two great circles passing through C1 and C2 and parallel to O1 and O2 (Fig 1B). 3) Determination of the elevation angle θ of P (invisible Sun) (Fig 1C). 4) In the knowledge of the position of P (i.e. θ and the azimuth direction of P), determination of the geographical north with the use of the sun-compass and a shadow-stick (Fig 1D).

In psychophysical laboratory and planetarium experiments, the error functions of these four steps of SPVN have been measured (1st step: [21], 2nd step: [22], 3rd step: [23], 4th step: [24]). Using full-sky imaging polarimetry, the patterns of the degree and direction of polarization of skylight have been measured as functions of the solar elevation angle θ and percentage ρ of cloud cover [25–31]. Combining these celestial polarization patterns with the error functions of the four steps of SPVN, the average north error NE of SPVN was determined as functions of θ and ρ for (a) spring equinox (21 March) and summer solstice (21 June), (b) cordierite, tourmaline and calcite sunstones, (c) morning and afternoon [24].

Based on the NE(θ, ρ) dataset, Száz and Horváth [32] demonstrated the high success of SPVN, that is the great probability Viking sailors could reach south Greenland from Norway along the 60° 21' 55" latitude using exclusively sky polarization. As input data, their computer simulations used numerous sky polarization patterns and the error functions of the four steps of SPVN.

Recently, Takács *et al.* [33] studied the sensitivity of the success of SPVN to the most relevant sailing, meteorological and navigation parameters, such as sunstone type, sailing date, navigation periodicity, night sailing, dominance of strongly, medium or weakly cloudy skies, and changeability of cloudiness. They showed that the navigation success is most sensitive to night sailing, navigation periodicity and sailing date, but robust against weather conditions. What were so far lacking in this experimental-computational archeological approach are the following:

- The study of the success of SVN and a combined navigation method using solar cues in sunshine (SVN) and sky polarization when the Sun is invisible (SPVN). Obviously, in sunshine it is not worth using SPVN instead of the much simpler SVN. Since during a several-week voyage both sunny and cloudy skies occur frequently, the use of the combined SVN-SPVN method is more realistic than the mere use of the four-step complex SPVN having four error functions contrary to the one-step SVN.

- The purpose of this work is to determine the success of the sole SVN and the combined SVN-SPVN relative to the mere SPVN. We investigated four different navigation methods:

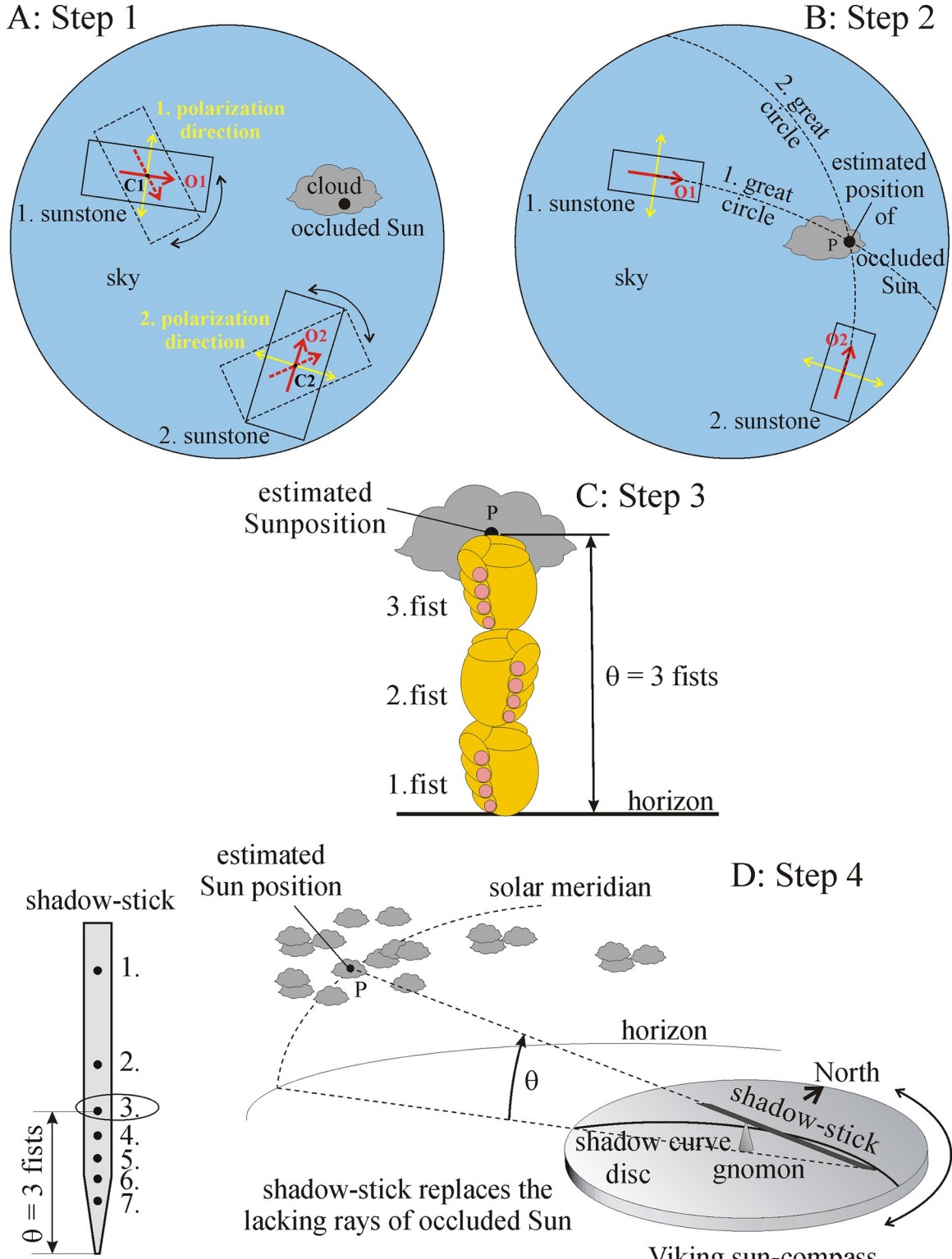

**Fig 1. The four steps of sky-polarimetric Viking navigation [18–20].** (A) In the 1. step, the Viking navigator looked at two points of the polarized sky through two sunstones. Rotating both sunstones, he determined the two directions pointing toward the Sun. These directions were scratched onto the sunstones as red arrows. (B) In the 2. step, he estimated the intersection of the two celestial great circles crossing the red arrows. This intersection coincides with the position of the occluded Sun. (C) In the 3. step, he measured the elevation θ (calculated in fist and finger) of the estimated position of the occluded Sun (θ = 3 fists + 0 finger in C). (D) In the 4. step, he

placed the hole (now 3.) of the shadow-stick corresponding to θ onto the tip of the vertical gnomon of the sun-compass while orienting the long axis of the shadow-stick toward the estimated position of the occluded Sun. He turned the horizontal disc of the sun-compass around the gnomon until the mirror-symmetric hyperbolic shadow curve scratched onto the disc hit the end of the shadow-stick resting on the disc. Then the symmetry axis of the shadow curve (marked by an arrow scratched into the disc) pointed toward the geographical North.

1) SPVN, using only sky polarization. 2) SVN or SPVN, using SVN in sunshine and SPVN under cloudy conditions. 3) SVN without stop, continuing sailing when the Sun is invisible. 4) SVN with stop, ceasing sailing when the Sun is invisible. The success of these four navigation methods was determined for three different navigator types (determining the intended sailing direction with large, medium or small frequencies) at spring equinox and summer solstice, with and without night sailing. We demonstrate here that if the navigators' aim is to maximize the sailing success, they have to choose different navigation methods depending on their navigation frequency.

- Compared to earlier studies that considered only SPVN [32,33], the uniqueness of our present work is that we investigate here not only SPVN, but also two further navigation methods using the sun-compass only in sunshine and continuing/ceasing sailing when the Sun is not visible. Furthermore, we also studied the fourth, combined method, which uses only the sun-compass when the Sun shines, while under a cloudy sky it analyses the sky polarization with a calcite sunstone in order to be able to use the sun-compass. Further novelty is that contrary to earlier studies, in this work we compare the navigation successes of often-, medium- and seldom-navigating sailors for nine different combinations of date (equinox, solstice) and night sailing (cease or continuation of sailing at night) versus the four navigation methods.

In the last few years, some new orientation methods have been developed which use polarized skylight for different navigation applications [34–40]. However, considering their instruments/equipments and how they use sky polarization, these sky-polarization-based modern navigation methods differ considerably from the SPVN.

## 2. Methods

### 2.1. Symbols and abbreviations used in this work

SVN: solar Viking navigation
  SPVN: sky-polarimetric Viking navigation
  NE: north error
  $N_{\text{successful}}$: number of successful Viking voyages
  $N_{\text{unsuccessful}}$: number of unsuccessful Viking voyages
  $S = N_{\text{successful}}/(N_{\text{successful}} + N_{\text{unsuccessful}})$: success ($0 \leq S \leq 1$) of Viking voyages
  okta: 1/8 = 12.5% of the full sky
  ρ: cloudiness of the sky (0 okta $\leq$ ρ $\leq$ 8 okta)
  $\Delta t$: navigation periodicity measured in hour
  $m$: dominance of cloudiness ($0 \leq m \leq 1$)
  σ: changeability of cloudiness measured in okta
  δ: directional drift of Viking ship measured in degree (˚)
  Δδ: deviation of directional drift measured in degree (˚)
  α: angle of the ship's velocity vector measured in degree (˚)
  $\underline{v}$: velocity vector of the ship
  $d$: visibility distance of the coastline of Greenland

*p*: chance/probability of invisible Sun ($0 \leq p \leq 1$)
Eq: spring equinox (21 March)
So: summer solstice (21 June)
+Ns: with night sailing (ship continues its course at night)
-Ns: without night sailing (ship stops at night)

## 2.2. Novelties of simulations

In our present simulations, we improved our earlier approach of simulated voyages using SPVN [33]. Fig 2 demonstrates qualitatively our developed computation/simulation method. Here we describe in detail only the new or most important issues of simulation, especially the four different navigation methods Viking sailors could apply. Our main aim was to find the best method ensuring the highest navigation success *S*. The simulated Viking voyages from Norway to Greenland ran along the 60° 21' 55" latitude. The parameters of simulations were the sailing date (spring equinox: 21 March, or summer solstice: 21 June, night sailing (sailing or not sailing at night), solar elevation, ship's speed, start time of diurnal sailing, north error, visibility distance of Greenland's southern coastline, navigation periodicity ($0.5\,\text{h} \leq \Delta t \leq 6\,\text{h}$), dominance ($-1 \leq m \leq 1$) and changeability ($1 \leq \sigma \leq 4$) of cloudiness. The settings of these parameters and all other details of the simulation of Viking voyages are described in [33].

## 2.3. Navigation methods

The simulations were done according to the following four navigation methods:

- **Method 1 (SPVN: only sky polarization)**: The navigator used SPVN, that is a calcite sunstone to determine the intended westward sailing direction on the basis of sky polarization. This navigation method was composed of the 4 steps of SPVN resulting in the cumulation of the north errors of these steps.

- **Method 2 (SVN or SPVN)**: The navigator used a calcite sunstone only when the Sun was invisible (occluded by clouds), which was calculated in the following way: At a given cloudiness ρ (0 okta: no cloud = visible Sun, 8 okta: total overcast = invisible Sun), a random number was generated from a uniform distribution, and the chance of clouded Sun increased linearly with ρ. If in the simulated weather situation the Sun was visible, then only the north errors of the 4th step of SPVN were used. If the Sun was invisible, then all 4 steps of SPVN were performed resulting in the cumulation of the north errors of them. This method is a more realistic approach than SPVN, because the navigator does not need sky polarization when the Sun is visible.

- **Method 3 (SVN without stop)**: The navigator did not have a sunstone (did not use sky polarization), so navigation (determination of the intended westward sailing direction) happened only when the Sun was visible (calculated with the same method as described at Method 2), and the sailing continued in the last-determined direction, if the Sun was invisible due to clouds. This navigation method used only step 4 of SPVN resulting in the north error of this step.

- **Method 4 (SVN with stop)**: The navigator did not use sky polarization, thus navigation was possible only when the Sun was visible (calculated with the same method as described at method 2), and the sailing stopped when the Sun was invisible. This method uses only step 4 of SPVN. Although it is also unknown what Vikings did when they did not see the Sun, the aim of the introduction of this method into our study is to determine the effect of ship's stop at occluded Sun on the success of SVN.

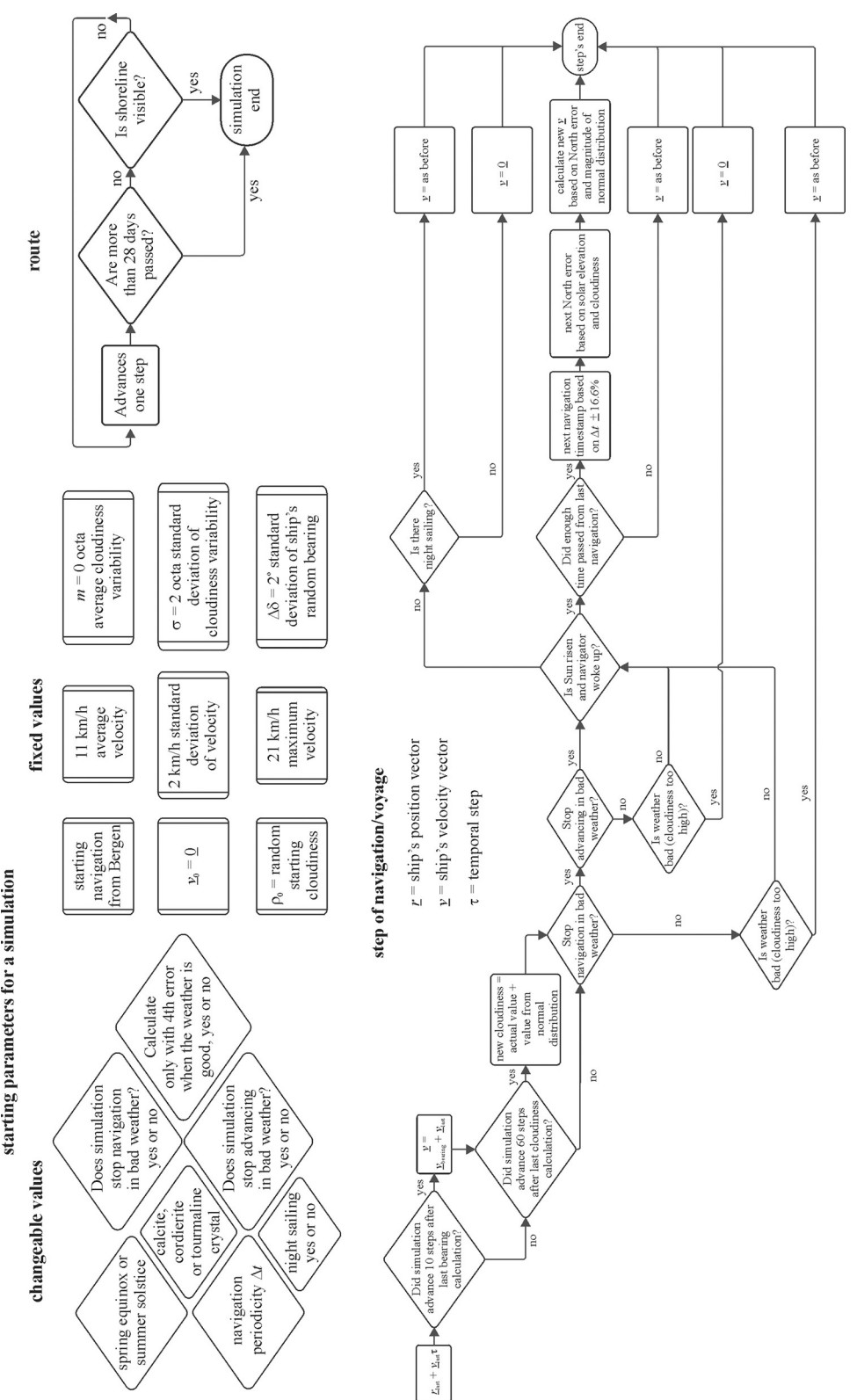

**Fig 2. Block diagram of our developed computation/simulation of Viking navigation.**

## 2.4. Ship's direction drift

The ship's direction drift was not simulated by Takács *et al.* [33]. In our new simulations it was necessary to take into account this drift, because using method 4 the ship might have been at a stand and drifted for several days in the sea waiting for cloudless weather. Due to sea waves, the actual motion direction of a ship varies randomly. Without such disturbing waves, the ship would not require navigation (i.e. periodical determination of the intended westward direction), because it would always be moving in the right (west) direction. We introduced the ship's directional drift δ in order to model the situation when the ship spends certain periods stopping in place at night or in the case of invisible (cloud-hidden) Sun. Before every simulation, we calculated the so-called drift's deviation $\Delta\delta$ so that the standard deviation $3° \leq \Delta\delta \leq 5°$ of a Gaussian distribution function was taken. This was modelled in such a way that in some simulations the average drift $\delta_{ave}$ was higher, while in others $\delta_{ave}$ was lower, in order to model realistic scenarios. During the simulation, every $\varepsilon = 10$ minutes δ was chosen randomly and uniformly from the interval $-\Delta\delta \leq \delta \leq +\Delta\delta$. In the *i*-th 10-minute period, we calculated the angle $\alpha_i = \arctan(v_{xi}/v_{yi})$ of the ship's velocity vector $\underline{v}(v_{xi} = \sin\alpha_i, v_{yi} = \cos\alpha_i)$, and determined the new angle $\alpha_{i+1} = \alpha_i + \delta$.

## 2.5. Visible or invisible (cloud-hidden) Sun

The cloudiness ρ is the proportion of clouds relative to the whole sky. We simulated whether or not the Sun is visible to a Viking navigatior as follows: the chance *p* of invisible Sun was calculated as $p = 0.125\rho$, and thus the chance of visible Sun was $1–0.125\rho$. The cases of cloudy sky (invisible Sun) or clear sky (visible Sun) were obtained with a binomial random generator.

 We randomly selected the sailing date, the night sailing and the navigation periodicity. Then, we ran the simulations with these selected variables for the four methods. All other parameters were constant throughout the simulations. Every simulated voyage was 3-week long, assuming that in this period Vikings reached their goal (Greenland), or after 3 weeks they died of starvation. Takács *et al.* [33] showed that the sunstone type (calcite, tourmaline or cordierite) has no significant effect on the sailing success. Therefore, when sky polarization was used in the simulations, we used only the north errors determined for the calcite sunstone [18,21,24]. We simulated 2 200 000 voyages altogether. The simulations and data evaluation have been performed with custom-developed Python scripts.

## 2.6. Calculation of navigation success for 9 combinations of date and night sailing

Combining differently the sailing date (spring equinox or summer solstice) and night sailing (stop or continue sailing at night), we calculated the navigation success *S* for the following nine different cases:

1. **Eq+Ns**: circa 550 000 routes at spring equinox (21 March) with night sailing

2. **So+Ns**: circa 550 000 routes at summer solstice (21 June) with night sailing

3. **Eq-Ns**: circa 550 000 routes at spring equinox without night sailing

4. **So-Ns**: circa 550 000 routes at summer solstice without night sailing

5. **+Ns = Eq+Ns & So+Ns**: circa 1 100 000 routes with night sailing

6. **-Ns = Eq-Ns & So-Ns**: circa 1 100 000 routes without night sailing

7. **Eq = Eq+Ns & Eq-Ns**: circa 1 100 000 routes at spring equinox

8. **So = So+Ns & So-Ns**: circa 1 100 000 routes at summer solstice

9. **All = Eq+Ns & So+Ns & Eq-Ns & So-Ns**: All the 2 200 000 routes

## 2.7. Navigation success

Navigation means the determination/estimation (with solar cues or sky polarization) of the intended west direction a Viking ship has to follow in order to reach South Greenland. A simulated Viking voyage was successful, if it reached the south tip or the southeast section of the coastline of Greenland within the visibility distance $d$ depending on the actual cloudiness $\rho$. The used linear function $d(\rho)$ [33] modelled how $\rho$ influences $d$, where $\rho$ changed stochastically. If a ship did not reach South Greenland in 3 weeks, or reached the northern coastline of Greenland or the northeast coastline of North-America, the voyage was unsuccessful. For the data evaluation we determined three realistic types of Viking navigator using the following three different intervals of navigation periodicity $\Delta t$:

- **Often-navigating Viking** determining the intended direction with $1\,h \leq \Delta t \leq 2\,h$ having an averaged success $S_{\Delta t\,=\,1\text{-}2h}$.

- **Medium-navigating Viking** determining the intended direction with $3\,h \leq \Delta t \leq 4\,h$ having an averaged success $S_{\Delta t\,=\,3\text{-}4h}$.

- **Seldom-navigating Viking** determining the intended direction with $5\,h \leq \Delta t \leq 6\,h$ having an averaged success $S_{\Delta t\,=\,5\text{-}6h}$.

For a given combination of date and night sailing, we simulated numerous (circa 550 000 or 1 100 00) voyages using a given navigation method. Among these voyages $N_{successful}$ were successful, and $N_{unsuccessful}$ were unsuccessful. Finally, we obtained the navigation success $S = N_{successful}/(N_{successful} + N_{unsuccessful})$ of the concerning method for the selected combination of date and night sailing. On the basis of the success $S$ of a given navigator type (often-, medium-, or seldom-navigating) using a given navigation method (1, 2, 3, 4), we determined the rank number $N$ (= I., II., II., IV.) of methods so that the higher the $S$, the smaller is $N$ (Tables 1–3).

A given simulation ended, if the ship reached the northeast coastline of North-America (being an unsuccessful sailing route) or the visibility distance $d$ of the southeast coastline of Greenland's south tip (meaning a successful sailing route), depending on the cloudiness value $\rho$ (measured in okta) as follows: $d(\rho) = -16.02875 \cdot \rho + 128.23$ km. This formula originates from [32,33]. Supposing that in overcast/foggy weather ($\rho = 8$) $d(\rho = 8) = 0$ km, the $d(\rho)$ function was calculated with a linear interpolation between $d(\rho = 8) = 0$ km and $d(\rho = 0) = 128.23$ km. The linear function $d(\rho)$ modelled how the cloudiness can influence the $d$ of the coastline, where $\rho$ changed stochastically.

The map of North Atlantics was drawn by our custom-developed software. The contours of islands, peninsulas and continents were manually digitalized from a map available as open-source data from http://www.gnuplotting.org/plotting-the-world-revisited/ (the raw data points of the contours can be freely downloaded in text format from: http://www.gnuplotting.org/data/world_10m.txt). These open-source data can be freely used without permission/licence. The coastline of Greenland (Fig 3) as the goal of Viking voyages was limited to its southeast section, rather than its full east borderline in [32]. If the ship reached shore on the northern coastline of Greenland (without Viking settlements), the voyage was considered unsuccessful.

## 3. Results

Fig 3 shows the sailing routes of 3×4×1000 = 12 000 simulated Viking voyages using the four different navigation methods for navigation periodicities $\Delta t$ = 1, 3 and 5 hours. Although in

Table 1. Success rank number *N* (I. = best with the largest navigation success *S*, II., III., IV. = worst with the smallest navigation success *S*) of the four (1, 2, 3, 4) navigation methods with averaged navigation success *S* in nine different cases for an often-navigating Viking determining the intended direction very frequently with periodicity $\Delta t$ = 1–2 hours.

| often-navigating Viking sailor | | | | |
|---|---|---|---|---|
| cases (1.,. . ., 9.) | number (1, 2, 3, 4) of navigation method (success *S*) | | | |
| success rank number *N* | I. | II. | III. | IV. |
| 1. Eq+Ns | 3 (0.795) | 2 (0.780) | 4 (0.669) | 1 (0.240) |
| 2. So+Ns | 2 (0.998) | 1 (0.998) | 3 (0.877) | 4 (0.697) |
| 3. Eq-Ns | 2 (0.997) | 1 (0.994) | 3 (0.778) | 4 (0.399) |
| 4. So-Ns | 2 (0.935) | 3 (0.916) | 1 (0.796) | 4 (0.586) |
| 5. +Ns (= Eq+Ns & So+Ns) | 2 (0.889) | 3 (0.836) | 4 (0.683) | 1 (0.618) |
| 6. -Ns (= Eq-Ns & So-Ns) | 2 (0.967) | 1 (0.896) | 3 (0.847) | 4 (0.492) |
| 7. Eq (= Eq+Ns & Eq-Ns) | 2 (0.890) | 3 (0.786) | 1 (0.619) | 4 (0.533) |
| 8. So (= So+Ns & So-Ns) | 2 (0.967) | 1 (0.897) | 3 (0.896) | 4 (0.641) |
| 9. All (= Eq+Ns & So+Ns & Eq-Ns & So-Ns) | 2 (0.928) | 3 (0.841) | 1 (0.757) | 4 (0.587) |

The higher the success *S*, the smaller the rank numeral (I., II., II., IV.). Eq: Spring equinox, So: Summer solstice, +Ns: Sailing at night, -Ns: Not sailing (stop) at night.

Fig 3 we visualized only 12 × 1000 routes for a given method and $\Delta t$ (map), our complete simulation involved 2 200 000 voyages altogether. It is clearly seen that the success *S* strongly depends both on the navigation periodicity and method. For different navigation periodicities, different navigation methods give the maximum success. Fig 3 demonstrates well our findings described below.

## 3.1. Dependence of sailing success on navigation method and periodicity

Fig 4 shows the success *S* of 2 200 000 (All = Eq+Ns & So+Ns & Eq-Ns & So-Ns) simulated Viking voyages as a function of the navigation periodicity $\Delta t$ for navigation methods 1–4. Since method 1 was the same as used by the simulated Viking voyages in the study of Takács *et al.* [33], this method serves as a basis for comparison further on. Generally, the shape of the success function $S(\Delta t)$ of methods 1 and 2 using sunstones (sky polarization) is very different

Table 2. As Table 1 for a medium-navigating Viking determining the intended direction frequently with periodicity $\Delta t$ = 3–4 hours.

| medium-navigating Viking sailor | | | | |
|---|---|---|---|---|
| cases (1.,. . ., 9.) | number (1, 2, 3, 4) of navigation method (success *S*) | | | |
| success rank number *N* | I. | II. | III. | IV. |
| 1. Eq+Ns | 2 (0.927) | 1 (0.822) | 3 (0.692) | 4 (0.648) |
| 2. So+Ns | 1 (0.990) | 2 (0.976) | 3 (0.784) | 4 (0.695) |
| 3. Eq-Ns | 2 (0.755) | 3 (0.619) | 4 (0.398) | 1 (0.366) |
| 4. So-Ns | 3 (0.811) | 2 (0.654) | 4 (0.593) | 1 (0.175) |
| 5. +Ns (= Eq+Ns & So+Ns) | 2 (0.952) | 1 (0.906) | 3 (0.737) | 4 (0.671) |
| 6. -Ns (= Eq-Ns & So-Ns) | 3 (0.715) | 2 (0.704) | 4 (0.495) | 1 (0.270) |
| 7. Eq (= Eq+Ns & Eq-Ns) | 2 (0.841) | 3 (0.655) | 1 (0.595) | 4 (0.523) |
| 8. So (= So+Ns & So-Ns) | 2 (0.815) | 3 (0.797) | 4 (0.644) | 1 (0.584) |
| 9. All (= Eq+Ns & So+Ns & Eq-Ns & So-Ns) | 2 (0.828) | 3 (0.726) | 1 (0.589) | 4 (0.584) |

**Table 3. As Table 1 for a seldom-navigating Viking determining the intended direction rarely with periodicity Δt = 5–6 hours.**

| cases (1.,. . ., 9.) | number (1, 2, 3, 4) of navigation method (success S) | | | |
|---|---|---|---|---|
| success rank number N | I. | II. | III. | IV. |
| **1. Eq+Ns** | 2 (0.901) | 1 (0.876) | 4 (0.644) | 3 (0.639) |
| **2. So+Ns** | 3 (0.719) | 4 (0.691) | 2 (0.568) | 1 (0.118) |
| **3. Eq-Ns** | 3 (0.522) | 2 (0.501) | 4 (0.387) | 1 (0.064) |
| **4. So-Ns** | 3 (0.743) | 4 (0.592) | 2 (0.303) | 1 (0.000) |
| **5. +Ns (= Eq+Ns & So+Ns)** | 2 (0.743) | 3 (0.679) | 4 (0.668) | 1 (0.496) |
| **6. -Ns (= Eq-Ns & So-Ns)** | 3 (0.632) | 4 (0.489) | 2 (0.402) | 1 (0.032) |
| **7. Eq (= Eq+Ns & Eq-Ns)** | 2 (0.701) | 3 (0.580) | 4 (0.516) | 1 (0.471) |
| **8. So (= So+Ns & So-Ns)** | 3 (0.731) | 4 (0.642) | 2 (0.437) | 1 (0.060) |
| **9. All (= Eq+Ns & So+Ns & Eq-Ns & So-Ns)** | 3 (0.656) | 4 (0.579) | 2 (0.569) | 1 (0.266) |

from that of methods 3 and 4 without sunstones. The success of navigation methods depends strongly on the navigation periodicity Δt making it impossible to select the best method which should be used throughout all voyages. Therefore in the rest of the paper, we will present the averaged success S of the often-, medium- and seldom-navigating Viking sailors.

For all the 2 200 000 (All = Eq+Ns & So+Ns & Eq-Ns & So-Ns) simulated Viking voyages we found that both the often- and medium-navigating sailors are most successful (rank N = I.) if using method 2, and least successful (rank N = IV.) if using method 4. The seldom-navigating sailor is most successful (N = I.) with method 3 and least successful (N = IV.) with method 1. The rank numbers N and averaged successes S for all methods can be found in Tables 1–3.

## 3.2. Effect of night sailing

Figs 5 and 6 present the success S of circa 1 100 000 simulated Viking voyages with night sailing (+Ns = Eq+Ns & So+Ns) and without night sailing (-Ns = Eq-Ns & So-Ns) as a function of the navigation periodicity Δt for navigation methods 1–4. The success S of a given method varied when night sailing was or was not used. For the circa 1 100 000 simulated Viking voyages without night sailing (-Ns = Eq-Ns & So-Ns) we obtained (Tables 1–3) that the often-navigating sailor is most (I.) and least (IV.) successful if using method 2 and 4, respectively. On the other hand, both the medium- and seldom-navigating sailors are most (I.) and least (IV.) successful when they use method 3 and 1, respectively.

In the case of the circa 1 100 000 simulated Viking voyages with night sailing (+Ns = Eq+Ns & So+Ns) we found (Tables 1–3) that method 2 ensures the highest navigation success (rank I.) for all three navigator types. Both the often- and seldom-navigating sailors are the least successful (IV.) when using method 1, while the medium-navigating sailor has the lowest success (IV.) when using method 4.

## 3.3. Effect of sailing date

Figs 7 and 8 display the success S of circa 1 100 000 simulated Viking voyages at summer solstice (21 June) and spring equinox (21 March) as a function of the navigation periodicity Δt for navigation methods 1–4. Similar tendency can be observed as in Fig 4: the success S of methods 2, 3 and 4 are higher for higher Δt-values. Interestingly, method 1 has higher S for higher Δt at spring equinox than at summer solstice, while methods 2–4 generally have higher S at summer solstice. In the case of circa 1 100 000 simulated Viking voyages at summer

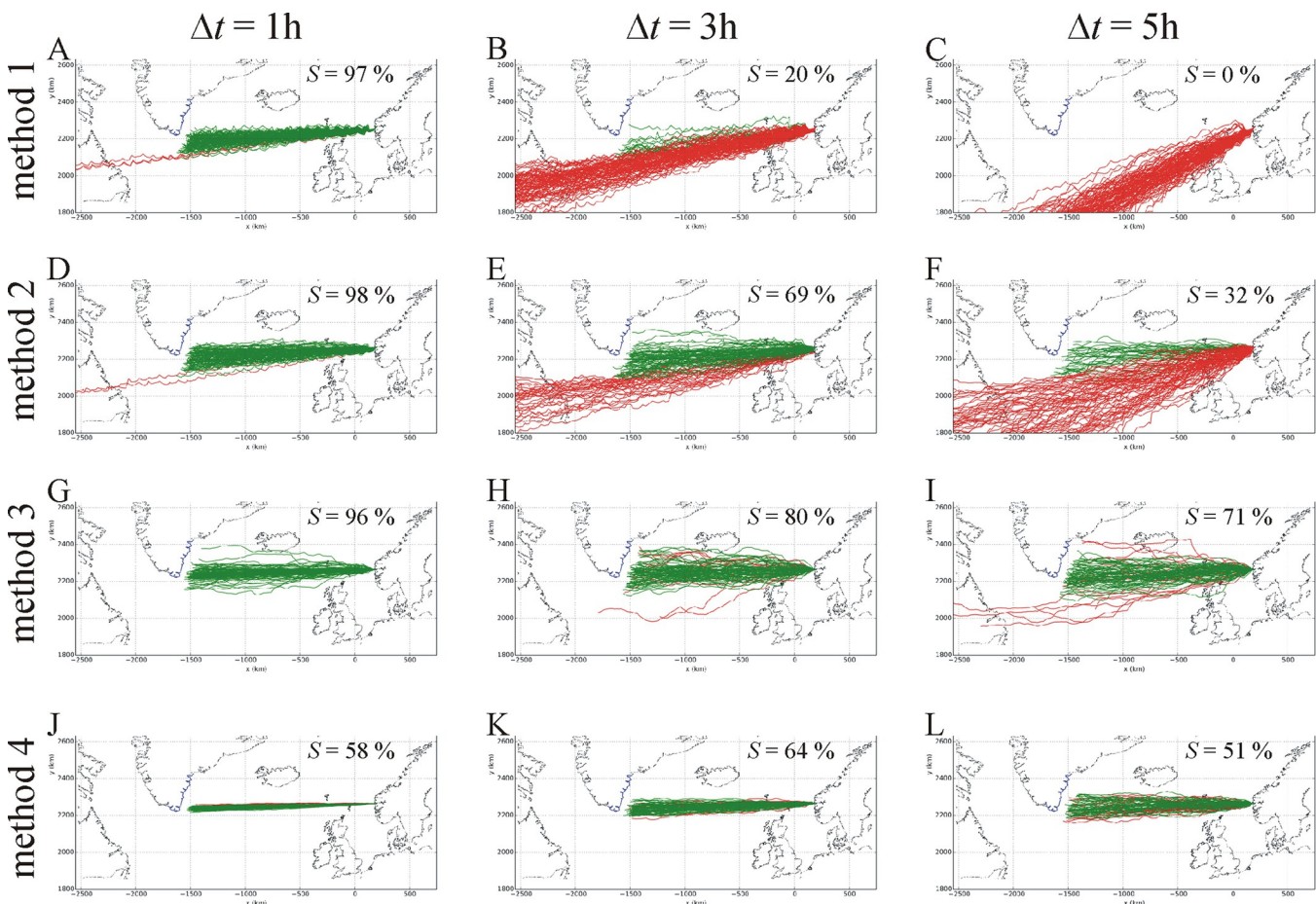

**Fig 3. Simulated sailing routes of Viking voyages without night sailing at summer solstice (21 June) using calcite sunstones between Norway and Greenland for navigation periodicities Δt = 1, 3 and 5 hours for four navigation methods with navigation success S (%).** The cloudiness dominance was $m = 0$, the cloudiness changeability was $\sigma = 2$, and the Brownian parameter for simulating the sea current effect was set to 4. The successful (reaching Greenland) or unsuccessful (not reaching Greenland) routes are marked with green and red, respectively. Every map demonstrates 1000 routes, the navigation success S of which is given in %. The values of the navigation success S are given in the maps. The blue curve is the borderline of visibility from which the southeast mountains of Greenland can already be seen from a Viking ship. The simulation of a voyage stops when the navigator sees the southeast coastline where the visibility distance is determined by the current cloudiness value ρ. Some simulated sailing trajectories pass through Iceland and/or North Scotland. In these cases, it was assumed that the Vikings continued their voyage towards Greenland. The contours of islands and continents are drawn by our custom-developed software after manual selection of contour points from open-source maps (from http://www.gnuplotting.org/plotting-the-world-revisited/).

solstice (So = So+Ns & So-Ns) we obtained (Fig 7, Tables 1–3) that method 2 results in the highest success (I.) for often- and medium-navigating sailors, and method 3 is the most successful (I.) for the seldom-navigating sailor. Method 1 ensures the lowest success (IV.) for medium- and seldom-navigating sailors, while method 4 is the least successful (IV.) for the often-navigating sailor.

For the circa 1 100 000 simulated Viking voyages at spring equinox (Eq = Eq+Ns & Eq-Ns) we found (Fig 8, Tables 1–3) that method 2 ensures the highest success (I.) for all three types of navigator, while method 4 is the least successful (IV.) for often- and medium-navigating sailors and method 1 for the seldom-navigating sailor.

### 3.4. Effect of date and night sailing

Figs 9 and 10 depict the success S of circa 550 000 simulated Viking voyages at summer solstice and spring equinox without night sailing. At summer solstice without night sailing (So-Ns) the

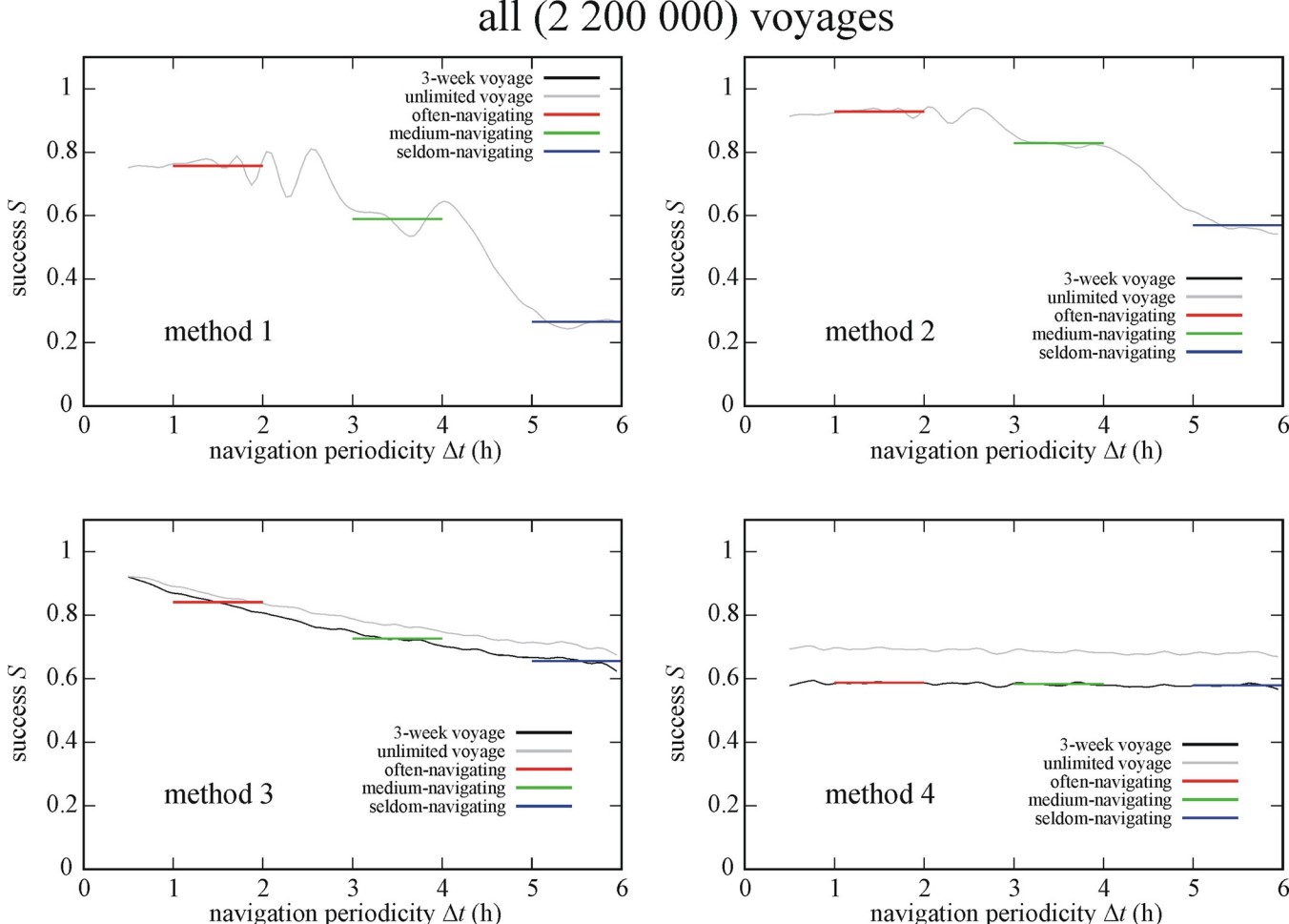

**Fig 4. Success $S$ of 2 200 000 (All = Eq+Ns & So+Ns & Eq-Ns & So-Ns) simulated Viking voyages as a function of the navigation periodicity $\Delta t$ for navigation methods 1–4.** The averaged successes $S_{\Delta t = 1\text{-}2h}$, $S_{\Delta t = 3\text{-}4h}$ and $S_{\Delta t = 5\text{-}6h}$ for often-, medium- and seldom-navigating sailors are marked with red, green and blue horizontal lines, respectively.

often-navigating sailor is the most (I.) and least (IV.) successful if using method 2 and 4, respectively (Fig 9). Medium- and seldom-navigating sailors are the most (I.) and least (IV.) successful if using method 3 and 1, respectively. At spring equinox without night sailing (Eq-Ns) method 2 ensures the highest success (I.) for often- and medium-navigating sailors, and method 3 for the seldom-navigating sailor (Fig 10). Method 1 results in the lowest success (IV.) for medium- and seldom-navigating sailors and method 4 for the often-navigating sailor (Tables 1–3).

Figs 11 and 12 show the success $S$ of circa 550 000 simulated Viking voyages at summer solstice and spring equinox with night sailing as a function of the navigation periodicity $\Delta t$ for navigation methods 1–4. At summer solstice with night sailing (So+Ns) method 2, 1 and 3 is the most successful (I.) for the often-, medium- and seldom-navigating sailor, respectively (Fig 11). On the other hand, method 4 is the least successful (IV.) for often- and medium-navigating sailors, and method 1 for the seldom-navigating sailor. At spring equinox with night sailing (Eq+Ns) method 2 is the most successful (I.) for medium- and seldom-navigating sailors and method 3 for the often-navigating sailor (Fig 12). Finally, method 1, 4 and 3 is the least successful (IV.) the often-, medium- and seldom-navigating sailor, respectively (Tables 1–3).

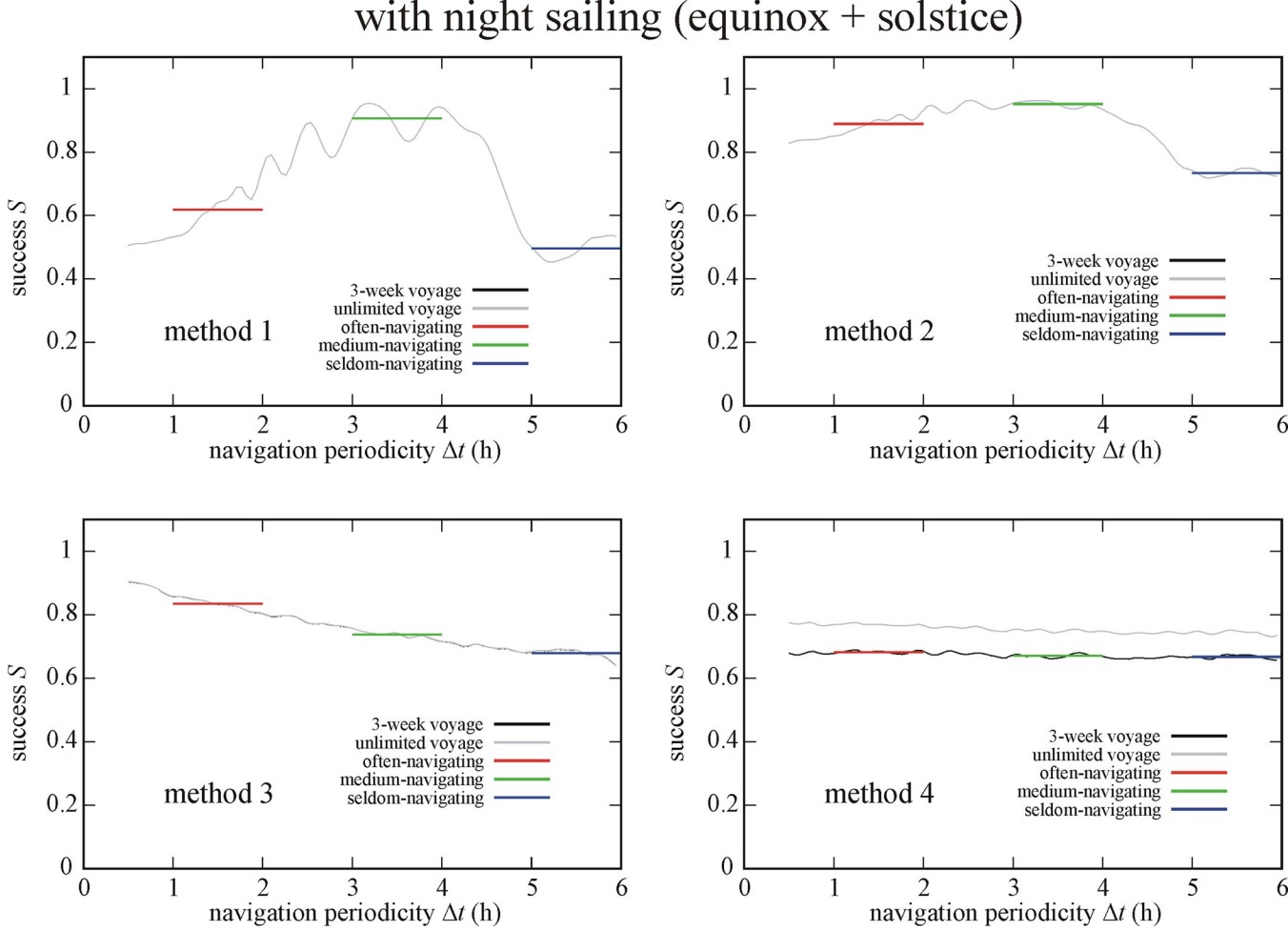

**Fig 5. Success $S$ of circa 1 100 000 (+Ns = Eq+Ns & So+Ns) simulated Viking voyages with night sailing as a function of the navigation periodicity $\Delta t$ for navigation methods 1–4.** The averaged successes $S_{\Delta t = 1\text{-}2h}$, $S_{\Delta t = 3\text{-}4h}$ and $S_{\Delta t = 5\text{-}6h}$ for often-, medium- and seldom-navigating sailors are marked with red, green and blue horizontal lines, respectively.

### 3.5. Comparison of Viking navigators

Table 4 compares the navigation success $S$ of the three navigator types in nine different cases versus the four navigation methods. It is clear from Table 4 that the order number i of sailors slightly depends on the case number and method. Summarizing the essence of Tables 4 and 5 shows the best navigator type having order number i. in Table 4 with the largest averaged navigation success $S$ in the 9 different cases studied versus the four navigation methods.

According to Tables 4 and 5, in cases 2. (So+Ns), 3. (Eq-Ns), 7. (Eq = Eq+Ns & Eq-Ns) and 9. (all = Eq+Ns & So+Ns & Eq-Ns & So-Ns), the often-navigating sailor is the best (with order number i) among the three navigator types for all four navigation methods. If using methods 1, 2 or 3, the navigator is the best in cases 4. (So-Ns), 6. (-Ns = Eq-Ns & So-Ns) and 8. (So = So +Ns & So-Ns). Using methods 3 and 4, the navigator is the best in cases 1. (Eq+Ns) and 5. (+Ns = Eq+Ns & So+Ns).

Using method 1, among the three navigator types the medium-navigating sailor is the best in case 5. (+Ns = Eq+Ns & So+Ns). Using method 2, the navigator is the best in cases 1. (Eq +Ns) and 5. (+Ns = Eq+Ns & So+Ns). Using method 3, the navigator is never the best. Using

## without night sailing (equinox + solstice)

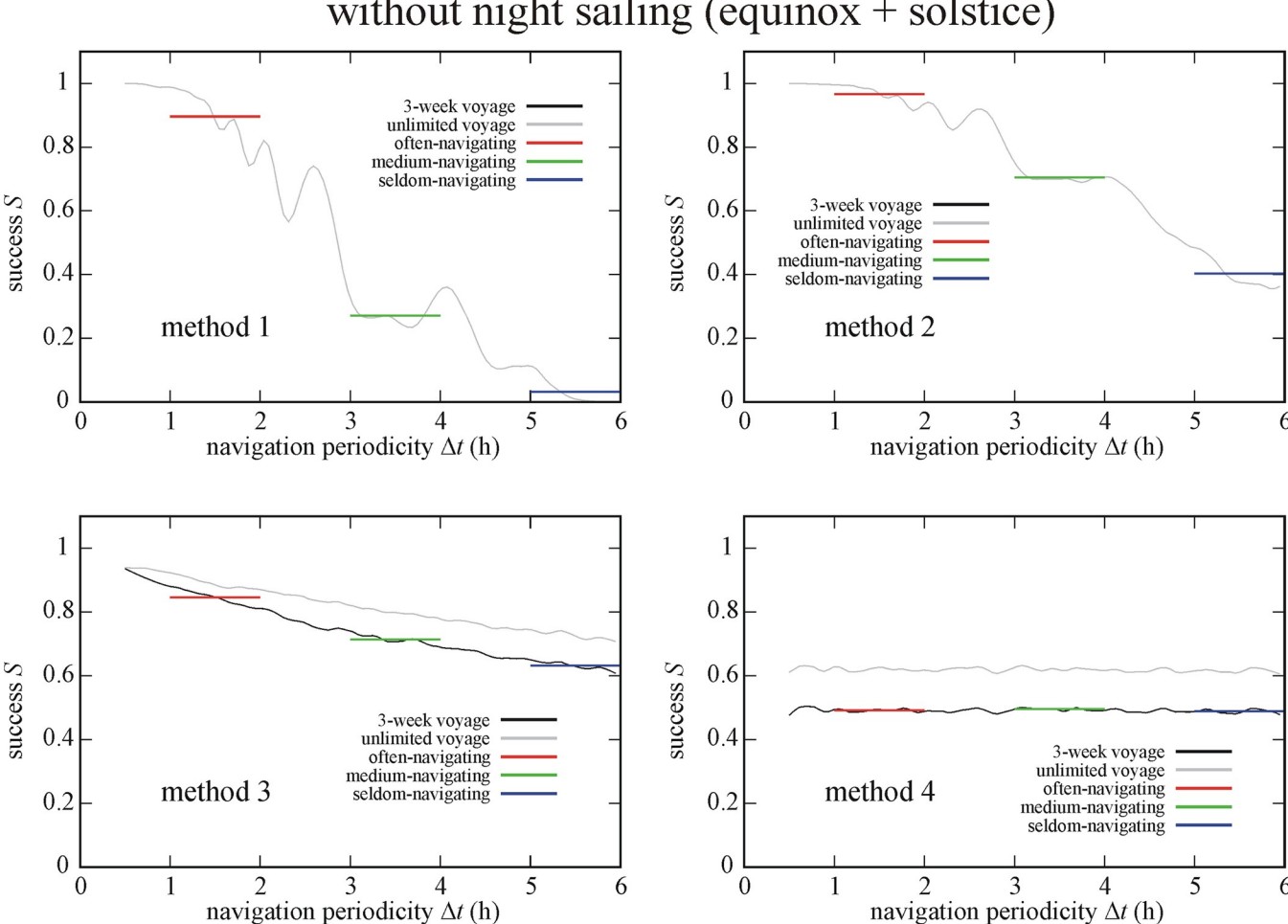

**Fig 6. Success *S* of circa 1 100 000 (-Ns = Eq-Ns & So-Ns) simulated Viking voyages without night sailing as a function of the navigation periodicity Δ*t* for navigation methods 1–4.** The averaged successes $S_{\Delta t = 1\text{-}2h}$, $S_{\Delta t = 3\text{-}4h}$ and $S_{\Delta t = 5\text{-}6h}$ for often-, medium- and seldom-navigating sailors are marked with red, green and blue horizontal lines, respectively.

method 4, the navigator is the best in cases 4. (So-Ns), 6. (-Ns = Eq-Ns & So-Ns) and 8. So (= So+Ns & So-Ns).

Among the three navigator types, the seldom-navigating sailor is the best with *S* = 0.876 only in case 1. (sailing at spring equinox day and night) when using only sky polarization (method 1: SPVN).

Hence, it is clear from Tables 4 and 5 that in most cases and for most navigation methods, the often-navigating sailor is the best (with order number i), that is this sailor type can navigate the ship with the highest sailing success.

## 4. Discussion

### 4.1. Viking navigation

Although the Sun and possibly the sky polarization were surely not the only tools for Viking navigation, the aim of this work is to estimate the success of four possible navigation methods using solely the Sun and/or sky polarization without other information. Therefore, our modelling disregarded a number of other factors of sail and also certain navigation traditions, such

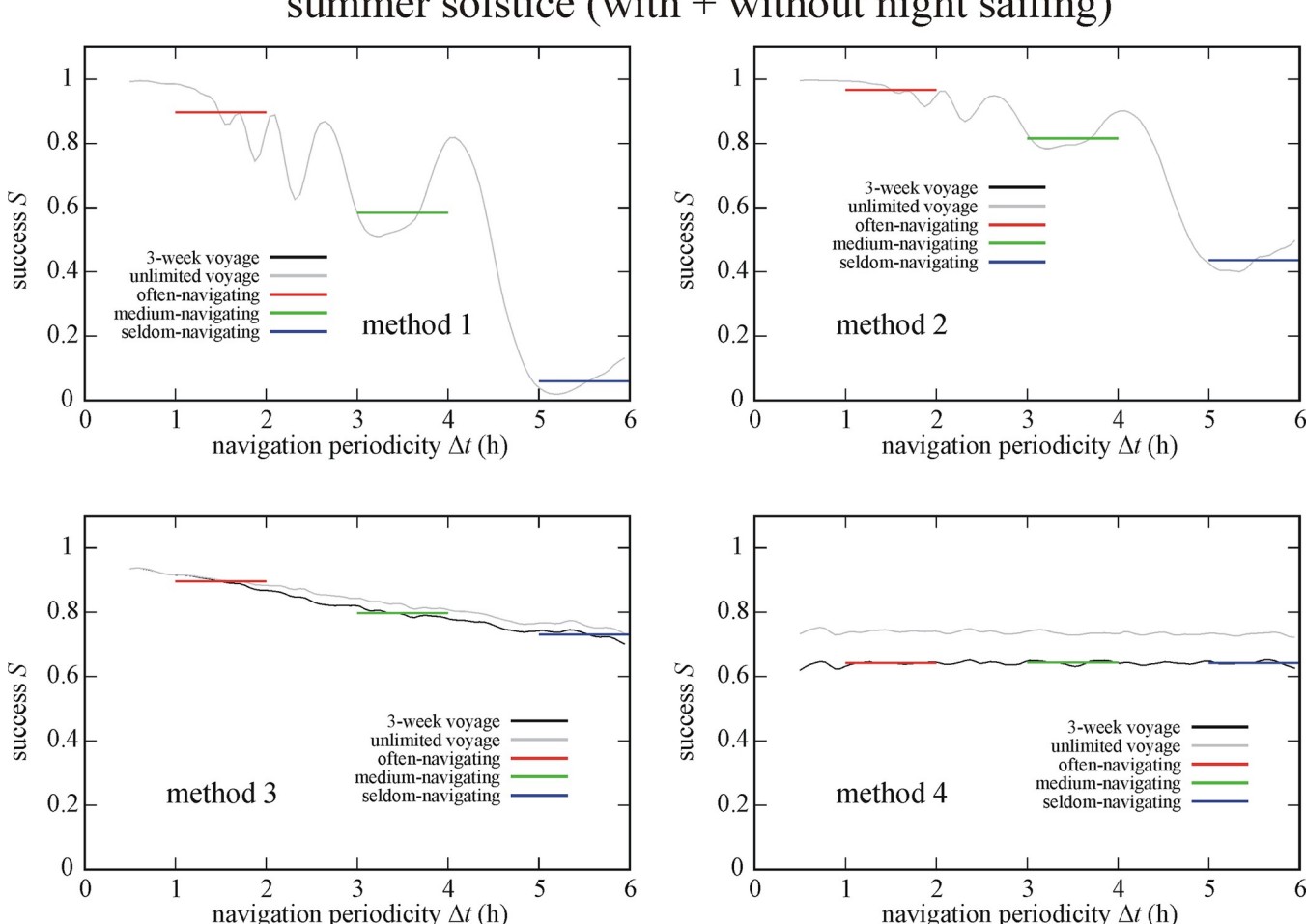

**Fig 7. Success _S_ of circa 1 100 000 (So = So+Ns & So-Ns) simulated Viking voyages at summer solstice (21 June) as a function of the navigation periodicity Δ_t_ for navigation methods 1–4.** The averaged successes $S_{\Delta t = 1\text{-}2h}$, $S_{\Delta t = 3\text{-}4h}$ and $S_{\Delta t = 5\text{-}6h}$ for often-, medium- and seldom-navigating sailors are marked with red, green and blue horizontal lines, respectively.

as following the migratory patterns of birds and whales, the prevailing and seasonal winds, stars and moon cycles and other means of navigating that are recorded with oral tradition and have also been tested experimentally and found to be effective [1–5,12,14].

- In one of the navigation methods of sailors crossing the Indian Ocean, at night the Polaris and/or the Moon served as time indicator (with the kind permission of the anonymous Reviewer 4) [41]:"*In this strategy, the lunar calendar was stringently followed where the movement of the moon in each constellation and zodiac sign was primarily taken into account. To determine the direction, some approximate calculations (Zam) were made, keeping before them the apparent motion of the Sun and North Star (Daru/Dhruv). Moreover, after knowing the angular distance of the Sun during noon and its shifts, the navigator could then calculate and note the latitude degrees in their logs accordingly. During the night the rising time of the North Star and the position of a cluster of other stars were important pointers for steering. It was known that there would be delays of some minutes at the time of the rise of stars. This information was used in gauging a particular direction. The units used for such celestial navigation was believed to follow cosmic laws and be attuned to energies of the universe. Being*

## spring equinox (with + without night sailing)

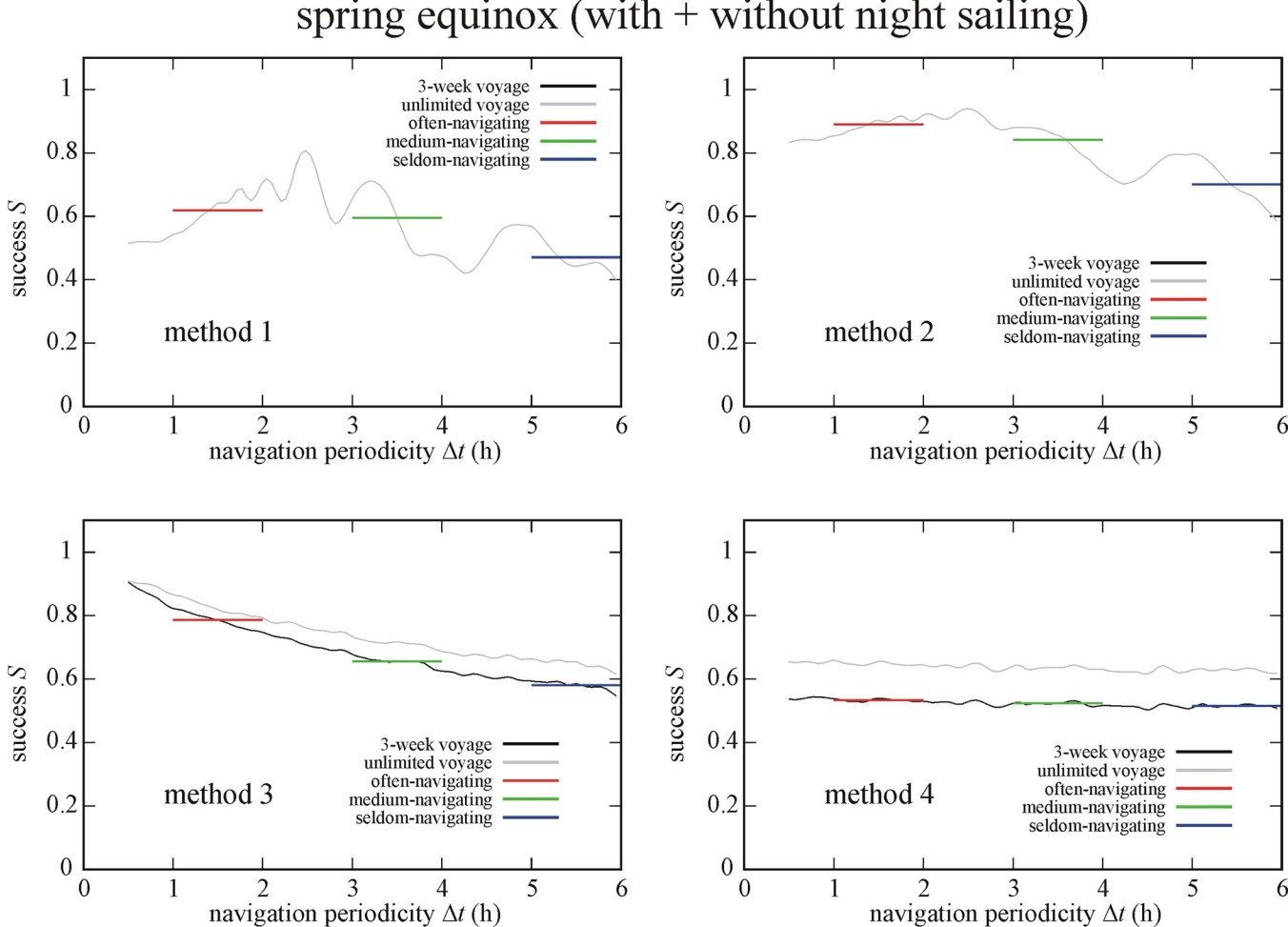

**Fig 8. Success S of circa 1 100 000 (Eq = Eq+Ns & Eq-Ns) simulated Viking voyages at spring equinox (21 March) as a function of the navigation periodicity Δt for navigation methods 1–4.** The averaged successes $S_{\Delta t = 1-2h}$, $S_{\Delta t = 3-4h}$ and $S_{\Delta t = 5-6h}$ for often-, medium- and seldom-navigating sailors are marked with red, green and blue horizontal lines, respectively.

guided by the monsoon winds the Malam (captain) was required to steer to sail north or south, as suitable, to bring Polaris to the altitude of the home port. Thereafter the vessel needed to steer the angular distance, or merely sail down the latitude, chiefly keeping Polaris at a constant angle. This calculation of distance was a key to the success of the voyage as the slightest mistake in calculation would mar the entire endeavor." Although this Indian maritime sailing knowledge of celestial navigation could have been very helpful also for Viking navigators, it is unknown whether they used any such or similar techniques during their night sailing. In principle, they could have simply calculated the latitude of Polaris as they progressed in the night sailing. However, unfortunately nobody knows how the Vikings steered in the night. Since we did not want to speculate on this, in our navigation model we assumed the simplest possibility of night sailing: at night the Viking ship either continued its course, or stopped until sunrise. In the latter case the average sailing speed is underestimated. The effect of this on the sailing success was, however, studied by comparison to the case when at night Vikings continued their course.

## summer solstice without night sailing

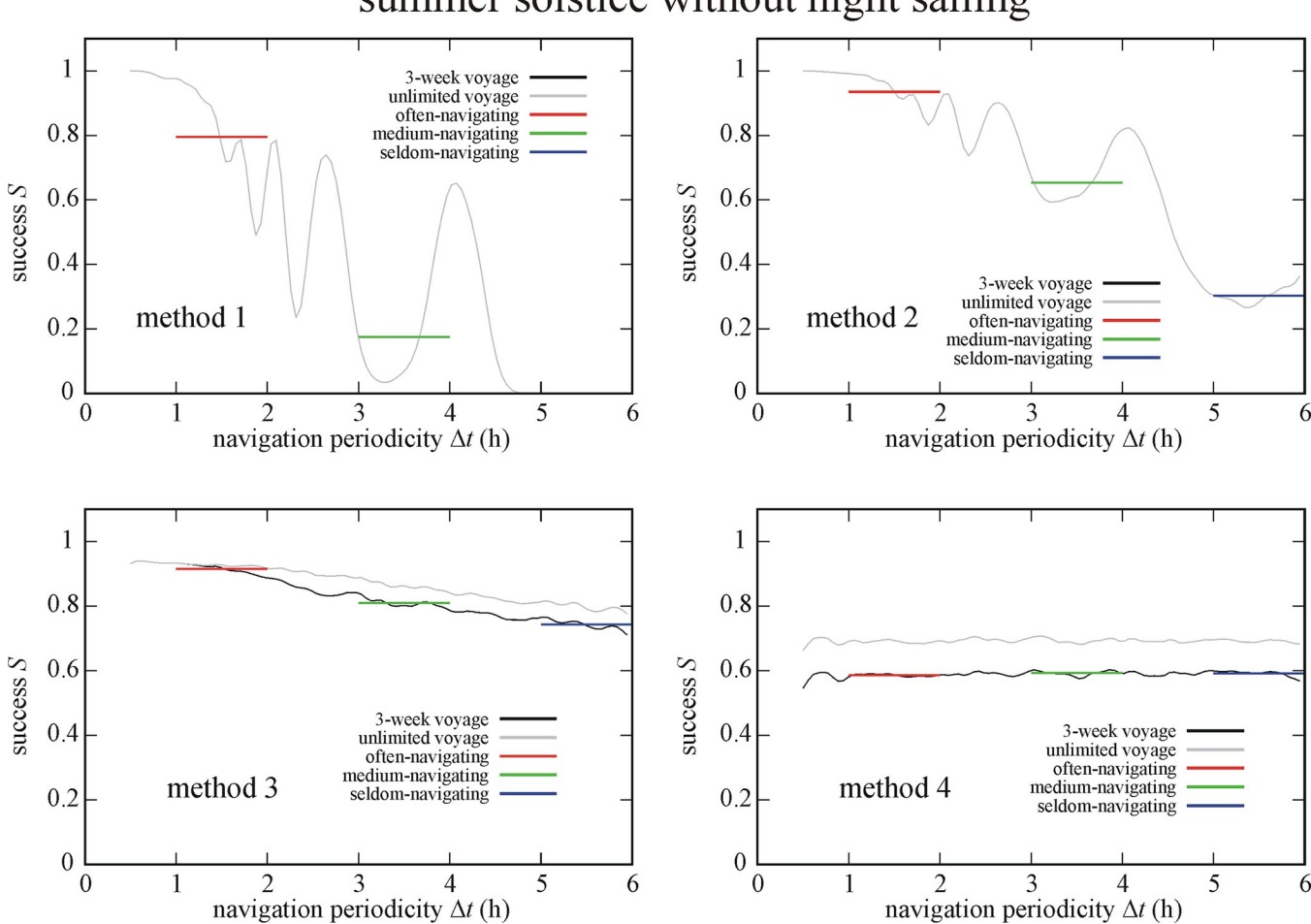

**Fig 9. Success *S* of circa 550 000 (So-Ns) simulated Viking voyages at summer solstice (21 June) without night sailing as a function of the navigation periodicity Δ*t* for navigation methods 1–4.** The averaged successes $S_{\Delta t = 1\text{-}2h}$, $S_{\Delta t = 3\text{-}4h}$ and $S_{\Delta t = 5\text{-}6h}$ for often-, medium- and seldom-navigating sailors are marked with red, green and blue horizontal lines, respectively.

- Bernáth *et al.* [42] proposed that the Viking sundial artefact may be an instrument to determine latitude and local noon. Although this hypothesis is supported by some quantitative data, the theory of solar (SVN) and sky-polarimetric (SPVN) Viking navigation based on the sun-compass artefact is a more widespread alternative interpretation confirmed by much psychophysical, meteorological, sky polarimetric and computational results [18–33]. While Takács *et al.* [33] investigated the robustness of the success *S* for all relevant navigation parameters, in this work we studied the effect of the navigation method only on the following parameters that influenced *S* significantly: navigation periodicity, sailing date and night sailing.

- Although in Fig 3J, 3K and 3L the success is *S* = 58, 64, 51%, respectively, the ratios of the numbers of green (successful) and red (unsuccessful) voyage trajectories do not reflect well these numerical values: much more green trajectories are visible than red ones. The simple reason for this apparent discrepancy is that many green trajectories overlap with red ones, and since the formers have been drawn over the latters, the green trajectories hide the underlying red ones. The opposite would be the situation, if the red trajectories were drawn over

## spring equinox without night sailing

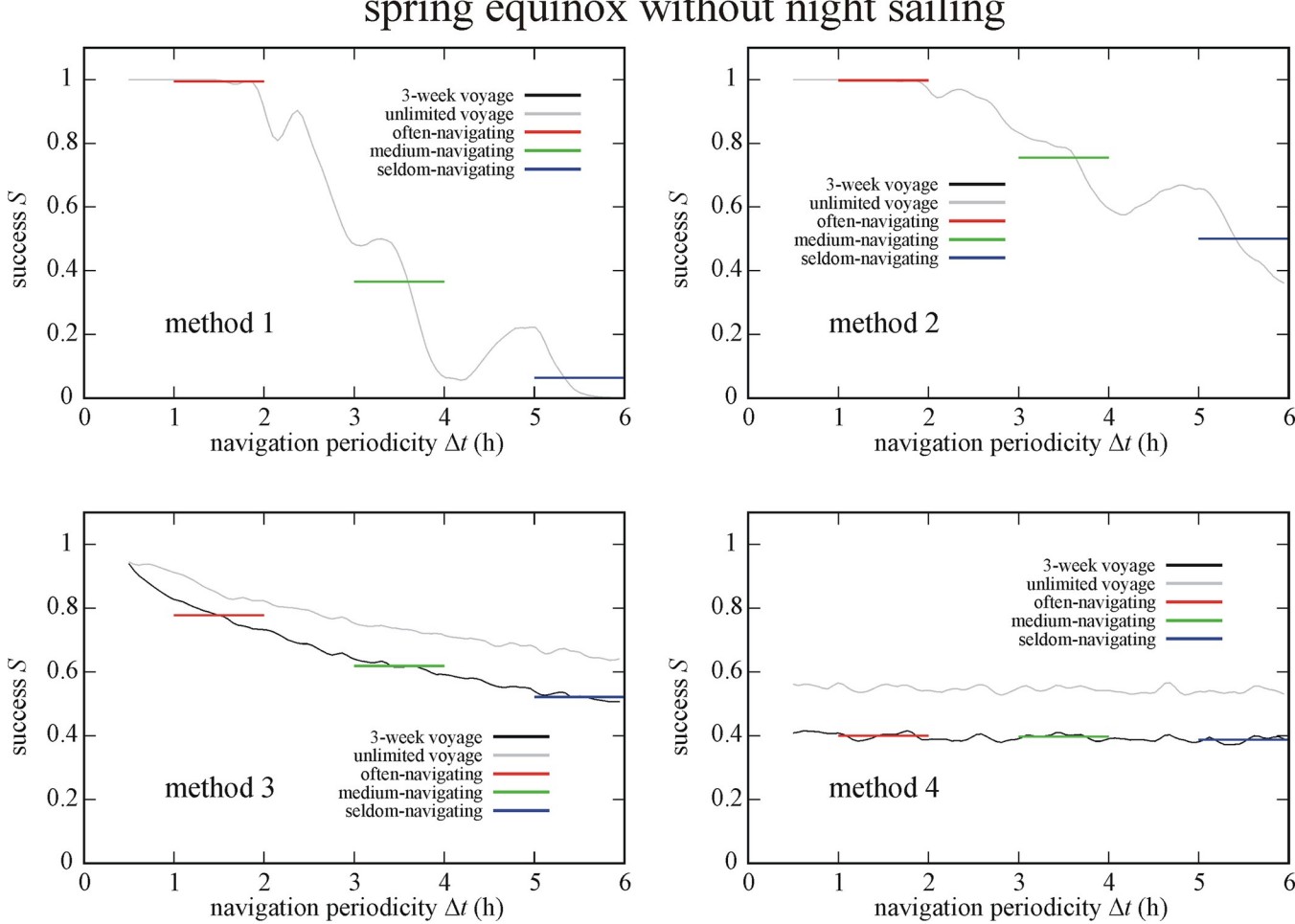

**Fig 10. Success $S$ of circa 550 000 (Eq-Ns) simulated Viking voyages at spring equinox (21 March) without night sailing as a function of the navigation periodicity $\Delta t$ for navigation methods 1–4.** The averaged successes $S_{\Delta t = 1\text{-}2h}$, $S_{\Delta t = 3\text{-}4h}$ and $S_{\Delta t = 5\text{-}6h}$ for often-, medium- and seldom-navigating sailors are marked with red, green and blue horizontal lines, respectively.

the green ones: then the red trajectories would hide the underlying green ones. Although this visualization problem cannot be eliminated, it is disturbing only if the success nears 50% ($S \approx 50\%$).

- We found that the success $S$ of Viking navigators ranked by the order number (i, ii, iii) slightly depends on the case (1.-9.) and the navigation method (1–4). This dependence is the result of the complex synergistic interaction of the sailing (sailing date, night sailing, directional drift $\delta$, ship's velocity $v$), meteorological (cloudiness $\rho$, dominance $m$ and changeability $\sigma$ of cloudiness, visibility distance $d$, probability $p$ of visible/invisible Sun) and navigation (sunstone type, navigation periodicity $\Delta t$, navigation method) variables/parameters of the solar (SVN) and the sky-polarimetric (SPVN) Viking navigation.

- We admit that the method of SPVN is rather complex, and nobody knows whether it has been really used by Viking sailors. Its practicality onboard a moving-waggling ship could be questioned, especially under rainy and/or windy conditions. The effects of these disturbing environmental factors on the success $S$ of Viking navigation are unknown and practically cannot be determined. It is, however, sure that they more or less reduce the $S$.

## summer solstice with night sailing

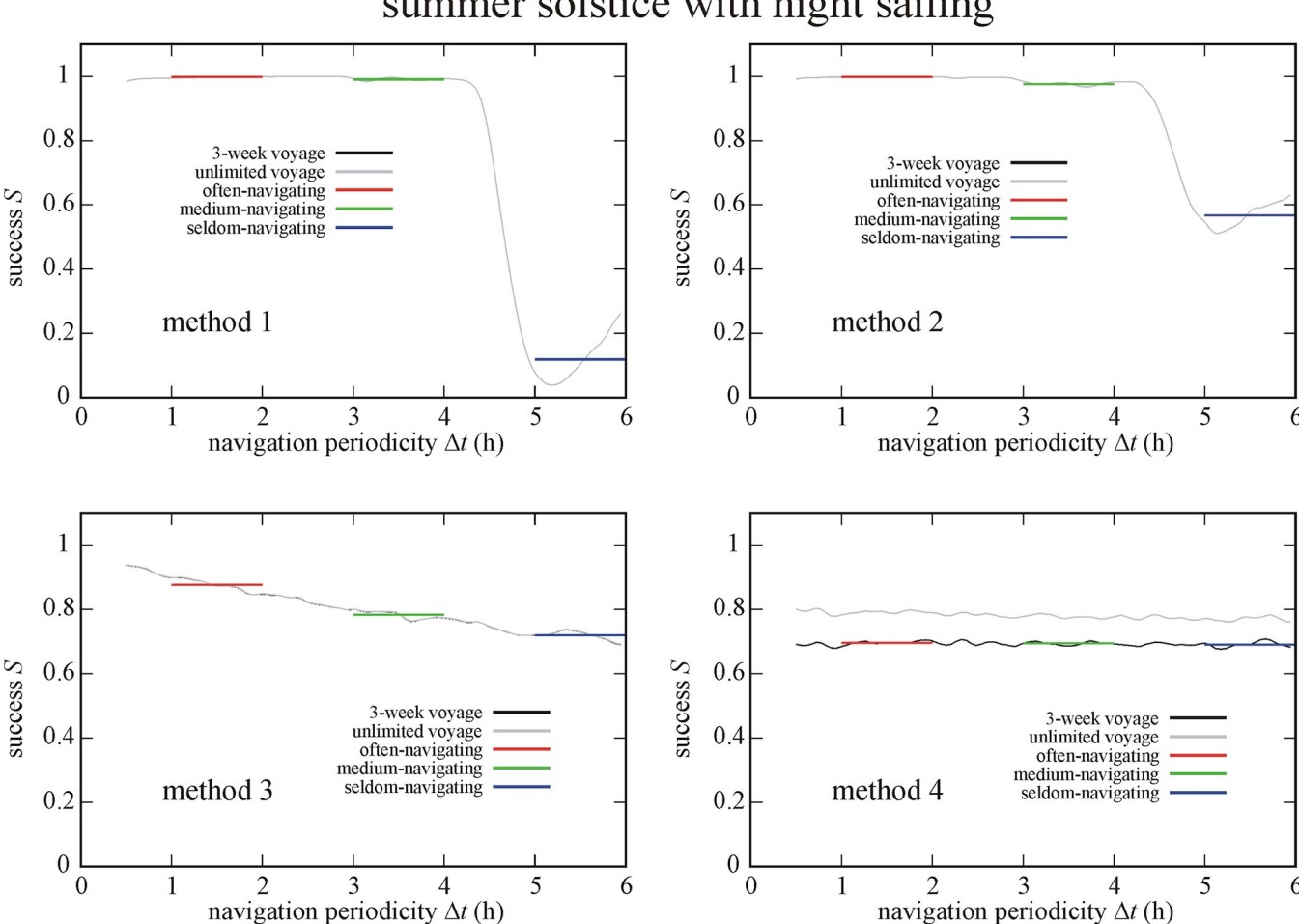

**Fig 11. Success S of circa 550 000 (So+Ns) simulated Viking voyages at summer solstice (21 June) with night sailing as a function of the navigation periodicity Δt for navigation methods 1–4.** The averaged successes $S_{\Delta t\,=\,1\text{-}2h}$, $S_{\Delta t\,=\,3\text{-}4h}$ and $S_{\Delta t\,=\,5\text{-}6h}$ for often-, medium- and seldom-navigating sailors are marked with red, green and blue horizontal lines, respectively.

### 4.2. Navigation method and periodicity

- Here we studied the problem that under which conditions and in which situations is it worth using solar (SVN) and/or sky-polarimetric (SPVN) Viking navigation if the goal is to maximize the sailing/navigation success *S*. We considered four navigation methods differing in the use of sunshine or sky polarization and in the stop or continuation of sailing at invisible Sun. *S* was also investigated as a function of the navigation periodicity Δt. Our main result is that for often-navigating sailors (with Δt = 1–2 h) it is worth using the Sun or sky polarization, while seldom-navigating ones (with Δt = 5–6 h) should rely on sunshine only in order to have maximal *S*.

- One could think that, among the four different methods, method 2 would always be triumphant due to its combined navigation technique (SPVN at invisible Sun, SVN at visible Sun), and because in sunshine its navigation error is smaller than for method 1. Száz *et al.* [32] and Takács *et al.* [33] have shown that in SPVN the navigation periodicity Δt is a crucial factor for the navigation success *S* due to the asymmetry of the navigation process (more navigation

## spring equinox with night sailing

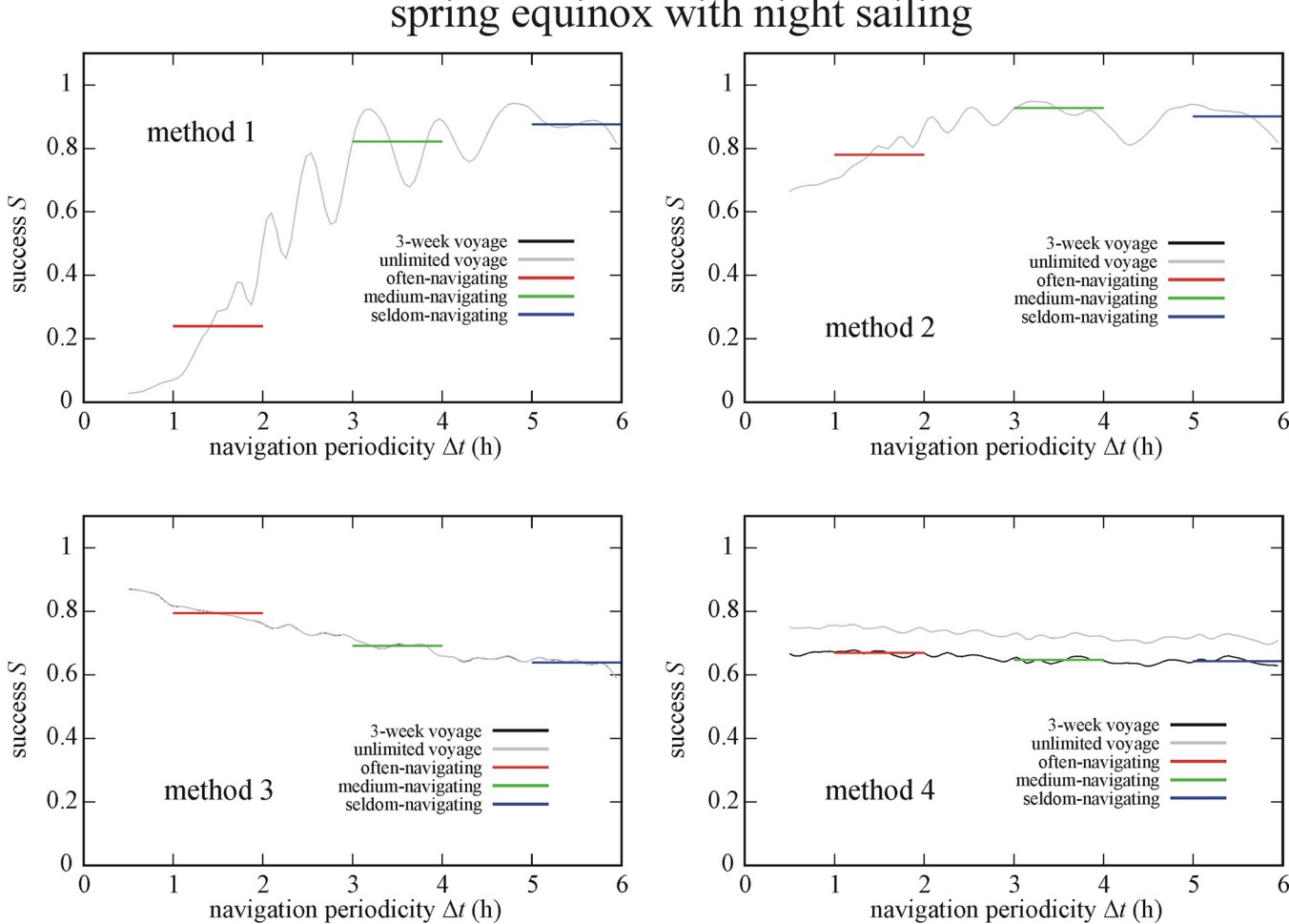

**Fig 12. Success $S$ of circa 550 000 (Eq+Ns) simulated Viking voyages at spring equinox (21 March) with night sailing as a function of the navigation periodicity $\Delta t$ for navigation methods 1–4.** The averaged successes $S_{\Delta t = 1\text{-}2h}$, $S_{\Delta t = 3\text{-}4h}$ and $S_{\Delta t = 5\text{-}6h}$ for often-, medium- and seldom-navigating sailors are marked with red, green and blue horizontal lines, respectively.

actions in the forenoon or the afternoon can result in high navigation error). The same is true for SVN, because it also uses the sun-compass to determine the intended sailing direction. Thus, it is logical that for larger $\Delta t$ (when the number of navigations are asymmetric) methods 1 and 2 have lower success ranks, and methods 3 and 4 (when there is no navigation in cases when there is a chance for inaccurate navigation) possess higher ranks.

### 4.3. Sailing date

Since the success of Viking voyages depends also on the sailing date (Tables 1–3, Figs 7–12), it is not all the same whether sailing happens at summer solstice (21 June) or spring equinox (21 March). If an often-navigating Viking sails at spring equinox, the length of the day is shorter than at summer solstice, giving a higher chance for the above-mentioned asymmetry, even if the navigation periodicity is 1 h $\leq \Delta t \leq$ 2 h. In this case, the best choice is to choose method 3, that is the Viking navigates only at visible Sun, can determine the intended sailing direction with a relatively high accuracy, and continues the voyage until the next chance for accurate navigation, when the Sun is visible again.

**Table 4. Order numbers i. (= best sailor with the largest averaged navigation success *S*), ii. (= second sailor with smaller *S*), iii. (= worst sailor with the smallest *S*) of the three Viking navigator types in nine different cases versus the four navigation methods.**

| cases (1.,. . ., 9.) | navigator type | order number (i, ii, iii) of navigators (success *S*) | | | |
|---|---|---|---|---|---|
| | | number (1, 2, 3, 4) of navigation method | | | |
| | | 1 | 2 | 3 | 4 |
| 1. Eq+Ns | often-navigating | iii. (0.240) | iii. (0.780) | **i. (0.795)** | **i. (0.669)** |
| | medium-navigating | ii. (0.822) | **i. (0.927)** | ii. (0.692) | ii. (0.648) |
| | seldom-navigating | **i. (0.876)** | ii. (0.901) | iii. (0.639) | iii. (0.644) |
| 2. So+Ns | often-navigating | **i. (0.998)** | **i. (0.998)** | **i. (0.877)** | **i. (0.697)** |
| | medium-navigating | ii. (0.990) | ii. (0.976) | ii. (0.784) | ii. (0.695) |
| | seldom-navigating | iii. (0.118) | iii. (0.568) | iii. (0.719) | iii. (0.691) |
| 3. Eq-Ns | often-navigating | **i. (0.994)** | **i. (0.997)** | **i. (0.778)** | **i. (0.399)** |
| | medium-navigating | ii. (0.366) | ii. (0.755) | ii. (0.619) | ii. (0.398) |
| | seldom-navigating | iii. (0.064) | iii. (0.501) | iii. (0.522) | iii. (0.387) |
| 4. So-Ns | often-navigating | **i. (0.796)** | **i. (0.935)** | **i. (0.916)** | iii. (0.586) |
| | medium-navigating | ii. (0.175) | ii. (0.654) | ii. (0.811) | **i. (0.593)** |
| | seldom-navigating | iii. (0.000) | iii. (0.303) | iii. (0.743) | ii. (0.592) |
| 5. +Ns (= Eq+Ns & So+Ns) | often-navigating | ii. (0.618) | ii. (0.889) | **i. (0.836)** | **i. (0.683)** |
| | medium-navigating | **i. (0.906)** | **i. (0.952)** | ii. (0.737) | ii. (0.671) |
| | seldom-navigating | iii. (0.496) | iii. (0.743) | iii. (0.679) | iii. (0.668) |
| 6. -Ns (= Eq-Ns & So-Ns) | often-navigating | **i. (0.896)** | **i. (0.967)** | **i. (0.847)** | ii. (0.492) |
| | medium-navigating | ii. (0.270) | ii. (0.704) | ii. (0.715) | **i. (0.495)** |
| | seldom-navigating | iii. (0.032) | iii. (0.402) | iii. (0.632) | iii. (0.489) |
| 7. Eq (= Eq+Ns & Eq-Ns) | often-navigating | **i. (0.619)** | **i. (0.890)** | **i. (0.786)** | **i. (0.533)** |
| | medium-navigating | ii. (0.595) | ii. (0.841) | ii. (0.655) | ii. (0.523) |
| | seldom-navigating | iii. (0.471) | iii. (0.701) | iii. (0.580) | iii. (0.516) |
| 8. So (= So+Ns & So-Ns) | often-navigating | **i. (0.897)** | **i. (0.967)** | **i. (0.896)** | iii. (0.641) |
| | medium-navigating | ii. (0.584) | ii. (0.815) | ii. (0.797) | **i. (0.644)** |
| | seldom-navigating | iii. (0.060) | iii. (0.437) | iii. (0.731) | ii. (0.642) |
| 9. All | often-navigating | **i. (0.757)** | **i. (0.928)** | **i. (0.841)** | **i. (0.587)** |
| | medium-navigating | ii. (0.589) | ii. (0.828) | ii. (0.726) | ii. (0.584) |
| | seldom-navigating | iii. (0.266) | iii. (0.569) | iii. (0.656) | iii. (0.579) |

The higher the success *S*, the smaller is the order number (i., ii., iii.). Eq: Spring equinox, So: Summer solstice, +Ns: Sailing at night, -Ns: Not sailing (stop) at night. The best order number i (and the *S*-value belonging to it) are in bold.

## 4.4. Night sailing

The sailing success *S* depends strongly also on the choice of night sailing (Tables 1–3, Figs 5, 6 and 9–12). Thus, it matters whether the sailing stops or continues at night. This dependence is expected, because at night the navigator cannot determine the position of the Sun, and thus cannot keep or correct the ship's course if it deviates from the intended westward sailing direction. Therefore, the disadvantage of night sailing is that the sailing success may decrease if night sailing is too frequently performed with incorrect courses. On the other hand, night sailing could also be advantageous, because it decreases the time in which the goal, Greenland is reached, if the correct course is kept at night. Stopping the ship at night can be advantageous, since it eliminates the possibly incorrect coursees and therefore increases the sailing success, but it may also be disadvantageous due to increasing the time of voyage. These advantages and disadvantages of night sailing interact synergistically with the sailing date, navigation method and periodicity and determine the success of Viking voyages.

**Table 5. The best navigator type (= having order number i. in Table 4 with the largest averaged navigation success $S$ in the 9 different cases studied versus the four navigation methods.**

| cases | navigation method | | | |
|---|---|---|---|---|
| | **1** | **2** | **3** | **4** |
| 1. Eq+Ns | seldom | medium | **often** | **often** |
| 2. So+Ns | **often** | **often** | **often** | **often** |
| 3. Eq-Ns | **often** | **often** | **often** | **often** |
| 4. So-Ns | **often** | **often** | **often** | medium |
| 5. +Ns (= Eq+Ns & So+Ns) | medium | medium | **often** | **often** |
| 6.–Ns (= Eq-Ns & So-Ns) | **often** | **often** | **often** | medium |
| 7. Eq (= Eq+Ns & Eq-Ns) | **often** | **often** | **often** | **often** |
| 8. So (= So+Ns & So-Ns) | **often** | **often** | **often** | medium |
| 9. All (= Eq+Ns & So+Ns & Eq-Ns & So-Ns) | **often** | **often** | **often** | **often** |

often: Often-navigating sailor with periodicity $\Delta t$ = 1–2 h. medium: Medium-navigating sailor with $\Delta t$ = 3–4 h. seldom: Seldom-navigating sailor with $\Delta t$ = 5–6 h. Eq: Spring equinox, So: Summer solstice, +Ns: Sailing at night, -Ns: Not sailing (stop) at night. The most frequent navigator type (**often**-navigating) occuring in this table is in bold.

## 5. Conclusions

From the results presented in this work we conclude the following:

If the navigation periodicity $\Delta t$ is between 1 and 2 hours (meaning an often-navigating Viking sailor), method 2 (SVN or SPVN) is the most successful in cases So+Ns, Eq-Ns, So-Ns, +Ns, -Ns, Eq, So and All, while method 3 (SVN without stop) has the largest success only in case Eq+Ns. (Table 1). Thus, for an often-navigating sailor, usually it is worth using sunstones (sky polarization).

If the navigation periodicity $\Delta t$ is between 3 and 4 hours (meaning a medium-navigating Viking), method 2 (SVN or SPVN) is the best in cases Eq+Ns, Eq-Ns, +Ns, Eq, So and All, but in cases So-Ns and -Ns method 3 (SVN without stop) is the most successful, while in case So +Ns method 1 (SPVN) has the highest success (Table 2). Thus, for a medium-navigating sailor, in some cases navigation without sunstones is better than navigating with them.

If the navigation periodicity $\Delta t$ is between 5 and 6 hours (meaning a seldom-navigating Viking sailor), method 3 (SVN without stop) is the most successful in cases So+Ns, Eq-Ns, So-Ns, -Ns, So and All, while in cases Eq+Ns, +Ns and Eq method 2 (SVN or SPVN) has the largest success (Table 3). Thus, for a seldom-navigating sailor, generally it is not worth using sunstones (sky polarization).

The often-navigating sailor has the highest average success $S$ (Tables 1–3), which implies that using sunstones (sky polarization) with very frequent navigation results in the best chance to survive the three-week-long voyage from Norway to Greenland.

In most cases and for most navigation methods, the often-navigating sailor can navigate the ship with the highest success, while the seldom-navigating sailor is the worst practically always with the lowest success.

As a future experimental continuation of this study, it would be worth investigating the accuracy of sky-polarimetric Viking navigation on the board of a vessel traveling along the 60° 21' 55" latitude between Norway and South Greenland under different sky conditions.

An important future work could be the experimental field test of the sailing success of different possible Viking navigation methods onboard a research vessel crossing the Atlantic Ocean between Bergen (Norway) and South Greenland.

## Acknowledgments

We thank four anonymous reviewers and Mir Muhammad Nizamani (one of the earlier reviewers), furthermore Uzair Aslam Bhatti, Hanna Landenmark and Zahid Mahmood Jehangiri (academic editors) for their valuable reviews and construcitve comments.

## Author Contributions

**Conceptualization:** Peter Takacs, Denes Szaz, Adam Pereszlenyi, Gabor Horvath.

**Data curation:** Peter Takacs, Denes Szaz, Gabor Horvath.

**Formal analysis:** Peter Takacs, Denes Szaz, Adam Pereszlenyi, Gabor Horvath.

**Funding acquisition:** Gabor Horvath.

**Investigation:** Peter Takacs, Denes Szaz, Adam Pereszlenyi, Gabor Horvath.

**Methodology:** Peter Takacs, Denes Szaz, Gabor Horvath.

**Project administration:** Gabor Horvath.

**Resources:** Gabor Horvath.

**Software:** Peter Takacs, Denes Szaz, Adam Pereszlenyi.

**Supervision:** Gabor Horvath.

**Validation:** Peter Takacs, Denes Szaz, Adam Pereszlenyi, Gabor Horvath.

**Visualization:** Peter Takacs, Denes Szaz, Gabor Horvath.

**Writing – original draft:** Peter Takacs, Denes Szaz, Adam Pereszlenyi, Gabor Horvath.

**Writing – review & editing:** Peter Takacs, Denes Szaz, Adam Pereszlenyi, Gabor Horvath.

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
