## [Decision Letter · Decision Letter 0]

14 Sep 2023

PONE-D-23-10265Sailing success of four Viking navigation strategies: often-navigating sailors can use the Sun or sky polarization, while seldom-navigating ones should rely on sunshine only in order to maximize the successPLOS ONE

Dear Dr. Horvath,

Thank you for submitting your manuscript to PLOS ONE. After careful consideration, we feel that it has merit but does not fully meet PLOS ONE’s publication criteria as it currently stands. Therefore, we invite you to submit a revised version of the manuscript that addresses the points raised during the review process.

ACADEMIC EDITOR:

In this round, the review was made by a total of 4 reviewers. Out of them Reviewers 1 to 3 are new reviewers. While, Reviewer 4 was the previous reviewer.  Seeing the reviews of the reviewer, I propose to authors to strictly follow the followings.

1. Please revise the title of this manuscript. It is very lengthy.

2. Authors must include a pseudo code of their developed method.

3. Authors must improve the resolution of all figures as per the reviewers concerns.

4. Authors must include a mathematical analysis in this manuscript.

5. Authors must improve the resolution of figures provided in the manuscript.

6. Authors must include the latest references in revised version of the manuscript.

We look forward to receiving your revised manuscript.

Kind regards,

Zahid Mahmood Jehangiri, Ph.D

Academic Editor

PLOS ONE

Journal Requirements:

Additional Editor Comments (if provided):

As per the reviewers comments, authors must do the following.

1. Please revise the title of this manuscript. It is very lengthy.

2. Authors must include a pseudocode of their developed method.

3. Authors must improve the resolution of all figures as per the reveiwers concerns.

4. Authors must include a mathematical analysis in this manuscript.

Reviewers' comments:

Reviewer's Responses to Questions

**Comments to the Author**

1. Is the manuscript technically sound, and do the data support the conclusions?

Reviewer #1: Partly

Reviewer #2: Yes

Reviewer #3: Yes

Reviewer #4: Yes

2. Has the statistical analysis been performed appropriately and rigorously? 

Reviewer #1: I Don't Know

Reviewer #2: N/A

Reviewer #3: Yes

Reviewer #4: Yes

3. Have the authors made all data underlying the findings in their manuscript fully available?

Reviewer #1: No

Reviewer #2: Yes

Reviewer #3: Yes

Reviewer #4: Yes

4. Is the manuscript presented in an intelligible fashion and written in standard English?

Reviewer #1: Yes

Reviewer #2: Yes

Reviewer #3: Yes

Reviewer #4: Yes

5. Review Comments to the Author

Reviewer #1: My comment is to improve the quality of the figures and article structure. Figures are blur. Also, the article is under review and it already contains the reviewers comments and response which is beyond my understanding. Moreover, too lengthy table captions are usually not good in research articles. Being out of my research portfolio and scope, I cannot fully understand this article.

Reviewer #2: My few suggestions are listed below.

1. Title of this manuscript is vague. Authors are advised to shorten the title.

2. Contributions are clear.

3. Authors should number each section. Currently, it is missing.

4. Please put a table for "Comparison of Viking navigators" and then explain. Current description is vague.

5. In Discussion section, please discuss your findings in the form of bullets. Currently, it is messed up.

6. In Conclusion, please remove all bullets and describe in a paragraph and also add one more sentence for future work.

7. In Methods, authors should include a pseudocode of their developed method. Currently, it is messed up and very hard for the reader to maintain flow. A pseudocode will give readers a fair insight about the developed algorithm.

8. Overall, novelty seems high and write up is good. Authors must adjust my above changes.

Reviewer #3: The authors have addressed all issues and comments raised by reviewer 3. Therefore, I don't have further comments.

The paper title is too long. This should be concisely described. I would also like to suggest improving figure quality. The reference are also quite old. only of the article is referred from 2022.

Reviewer #4: The authors have adequately revised their work as per the third and fourth reviewers' reports. Since the authors have requested the fourth reviewer to quote a paragraph on the Indian Ocean sailing techniques, they are given the permission to do so. The reference information on the said citation is as follows- Goswami, Chhaya. 2015 Malam Ni Pothi: Treatise on Kachchhi Navigation Techniques. Publisher- Darshak Itihas Nidhi, Mumbai. (Summarized transcription and a critique of the original sailing manuals written in old Gujarati.)

The authors have mentioned that it is little known that how exactly the Vikings sailed in the night. On this I suggest them to consider a separate investigation based upon the cutting edge field research, consulting fragmented unconventional archives (log book or any such written accounts), oral narratives, to trace out or guesstimate the patterns and rhythms of their lunar sailings.

I also have a suggestion for the authors regarding their long title, if they wish to change or modify accordingly.

"Speedy Bearings to Slacked Steering: Mapping the Navigational Patterns and Motions of the Viking Voyages"

Please note it is just a suggestion.

6. PLOS authors have the option to publish the peer review history of their article (what does this mean?). If published, this will include your full peer review and any attached files.

Reviewer #1: No

Reviewer #2: No

Reviewer #3: No

Reviewer #4: No

---

## [Author Response · Author response to Decision Letter 0]

10 Oct 2023

Our responses are uploaded separately: +VikingPolSunCloud_PLoS-One-response-3.docx

---

## [Editor Report · Decision Letter 1]

20 Oct 2023

Speedy Bearings to Slacked Steering: Mapping the Navigation Patterns and Motions of Viking Voyages

PONE-D-23-10265R1

Dear Dr. Horvath,

We’re pleased to inform you that your manuscript has been judged scientifically suitable for publication and will be formally accepted for publication once it meets all outstanding technical requirements.

Kind regards,

Zahid M. Jehangiri, Ph.D

Academic Editor

PLOS ONE

Additional Editor Comments (optional):

Since authors have addressed all the concerns that were raised by the reviewers in two rounds of this manuscript. Therefore, I am happy to accept this manuscript in its current form.
---

## [Editor Report · Acceptance letter]

30 Oct 2023

PONE-D-23-10265R1 

Speedy Bearings to Slacked Steering: Mapping the Navigation Patterns and Motions of Viking Voyages 

Dear Dr. Horvath:

I'm pleased to inform you that your manuscript has been deemed suitable for publication in PLOS ONE. Congratulations! Your manuscript is now with our production department. 

Kind regards, 

on behalf of

Dr. Zahid Mahmood Jehangiri 

Academic Editor

PLOS ONE